# Miniaturized Antenna Design for Wireless and Powerless Surface Acoustic Wave Temperature Sensors

**DOI:** 10.3390/s24175490

**Published:** 2024-08-24

**Authors:** Naranut Sreang, Jae-Young Chung

**Affiliations:** Department of Electrical and Information Engineering, Seoul National University of Science and Technology, Nowon-gu, Seoul 01811, Republic of Korea; naranut.sreang@gmail.com

**Keywords:** hemispherical helical antenna (HSHA), temperature sensor, surface acoustic wave (SAW), miniaturized antenna

## Abstract

This paper presents the introduction, design, and experimental validation of two small helical antennae. These antennae are a component of the surface acoustic wave (SAW) sensor interrogation system, which has been miniaturized to operate at 915 MHz and aims to improve the performance of wireless passive SAW temperature-sensing applications. The proposed antenna designs are the normal-mode cylindrical helical antenna (CHA) and the hemispherical helical antenna (HSHA); both designed structures are developed for the ISM band, which ranges from 902 MHz to 928 MHz. The antennae exhibit resonance at 915 MHz with an operational bandwidth of 30 MHz for the CHA and 22 MHz for the HSHA. A notch occurs in the operating band, caused by the characteristics of the SAW sensor. The presence of this notch is crucial for the temperature measurement by aiding in calculating the frequency shifting of that notch. The decrement in the resonance frequency of the SAW sensor is about 66.67 kHz for every 10 °C, which is obtained by conducting the temperature measurement of the system model across temperature environments ranging from 30 °C to 90 °C to validate the variation in system performance.

## 1. Introduction

Wireless temperature monitoring has emerged as a pivotal technology that has significantly enhanced operational efficiencies across multiple sectors through its capability for real-time remote observations. This technology has been widely used in several fields to assure adherence to essential requirements for the safety and effectiveness of products and services [1,2,3]. Healthcare is one of the primary sectors where wireless temperature monitoring is crucial. Monitoring the conditions within vaccine and drug storage facilities is crucial for preserving the integrity of pharmaceuticals. A specialized wireless health-monitoring system was developed to accurately measure internal body temperature [4,5]. The technology has a significant influence in the field of agriculture, specifically in monitoring greenhouse temperatures to enhance the optimal circumstances for plant growth, and a wireless temperature monitoring system using an infrared sensor was explored to ensure the consistent and accurate monitoring of plant canopy temperatures in agricultural environments [6]. Within the food sector, food quality and safety are ensured throughout the processing and storage stages. To facilitate this, a wireless energy-efficient temperature probe was built to monitor food processes. The system was designed to accurately monitor processes in low-temperature environments, specifically in pharmaceutical and food processing plants [7]. Furthermore, these systems play a vital role in building and home environments, energy sector, data centers, and aerospace engines [8,9,10].

A temperature sensor is an important part of a temperature-monitoring system. Various types of high- and low-temperature sensors, such as LC-based resonator sensors, radio frequency identification (RFID) sensors, resistance temperature detectors (RTDs) sensors, and wireless surface acoustic wave (SAW) sensors, have been studied and developed [11,12,13,14]. Among the temperature sensors, the SAW sensor has garnered increased interests in research due to its compact size, high sensitivity, rapid and fast response, and ability to function effectively in harsh environments, which have been consistently advancing, leading to improvements across all facets of the technology [15]. Several studies have shown that the SAW temperature device can be integrated with the antenna to perform well in challenging environments [16,17]. A SAW temperature sensor that utilizes a delay line was explicitly designed to function at 440 MHz and 2.4 GHz center frequencies, allowing it to effectively measure temperature in high-temperature settings [18]. The initial proof of concept system, which started at 250 MHz, progressed to the present at 915 MHz [19]. Most wireless passive SAW sensor systems are being developed and utilized at a frequency of 915 MHz for certain applications because of the advantages offered by the 915 MHz operating frequency ISM band. These advantages include less interference, extended communication range, and improved penetration compared to higher frequencies such as 2.4 GHz. This frequency is also considered optimal due to favorable device and compact antenna sizes and the widespread availability of compatible wireless RF components [20]. Wireless SAW sensors are designed to operate on 915 MHz; they have also been integrated with a disk monopole antenna with a size of 0.110λ and provide a broad band of 30 MHz [21]. A novel approach to temperature measurement using a wireless temperature sensor that depends on SAW technology was studied. Both the reader antenna, PIFA, and a normal-mode helical antenna were developed to operate at 915 MHz. The impedance matching network is required for efficient transmission, and the helical antenna size is about 0.070λ, which is relatively large [22].

Therefore, this work presents a miniaturized antenna design for this wireless passive and powerless SAW temperature sensor that operates at a central frequency of 915 MHz. We employed the High-Frequency Simulation Software (HFSS) for the simulations and optimization to achieve a compact size for two different types of normal-mode helical antennae: the non-uniform-pitch Cylindrical Helical Antenna (CHA) with a size of 0.055λ and the Hemispherical Helical Antenna (HSHA) with a size of 0.015λ. The antenna is mounted onto a coplanar waveguide (CPW), and the SAW sensor is directly installed next to the antenna without requiring a matching circuit. To verify the design, a low-temperature test was conducted to ensure optimal functionality of the system, as described in the following section.

## 2. Antenna Design

### 2.1. Non-Uniform-Pitch Cylindrical Helical Antenna (CHA) Design

The proposed system design configuration is composed of a surface acoustic wave sensor, which is a blue square box implemented on the shorted coplanar waveguide (CPW) structure with an FR-4 substrate material, a thickness of 1 mm, and a size of 24 × 24 mm^2^, as shown in Figure 1a. The top perspective view of the proposed system design is displayed in Figure 1b. When the antenna receives the signal from the reader, it transmits the electromagnetic wave to the coplanar waveguide (CPW) and SAW sensor. The CPW provides a low-loss pathway for the signal, ensuring efficient transmission. The SAW sensor, implemented on the CPW, interacts with the incoming electromagnetic wave. It is sensitive to changes in the surrounding environment, such as temperature fluctuations. When these alterations occur, they affect the properties of the acoustic wave produced within the sensor. The surface acoustic wave (SAW) sensor reflects the altered signal back through the CPW. The alterations in the reflected signal serve as an indication of the environment or temperature changes that have been noticed by the SAW sensor.

The essential factors to consider for an optimal helical antenna typically encompass helical length (l); pitch length (S), which is the distance between each successive coil of the helix; number of turns (N); the helix’s cross-section (A); and the most important parameter, which is the helix’s diameter (D). The equation below defines the relationship between these parameters and their influence on the antenna’s resonant frequency $\left(f_{c}\right)$ [23,24].
(1)$$f_{c}=\frac{\sqrt{S}}{2\pi D\sqrt{\pi \mu_{0}\mu_{r}CN}}$$
where C is the capacitance of the helical antenna system; based on the equation provided above, the length of the helix and pitch length has a proportional relationship with the resonant frequency. In contrast, the diameter and number of turns are inversely proportional to the resonant frequency of the helix. Designing the normal-mode helical antenna using a uniform pitch distance may result in a larger size. The incorporation and use of a helical antenna with a non-uniform pitch have been suggested in the design since it is a notable benefit by offering a bandwidth improvement of about 18% and offers a smaller size as described in [25].

The diameter (D) selection is of utmost importance when designing a helical antenna since it directly impacts its resonant frequency and overall performance. The diameter of the helix has to be significantly smaller than the wavelength to achieve this normal mode. Our proposed design for a normal-mode non-uniform-pitch helical antenna is chosen to be $D\approx \frac{1}{50}\lambda$. The antenna consists of 3 different pitch lengths as described in Figure 2a and Table 1.

A parametric study of the antenna diameter was conducted by varying the value from the initial value D=4mm to D=5.8mm. From Equation (Equation 1), the diameter of the helix and the resonance frequency are inversely proportional to each other, which matches the simulation study depicted in Figure 2b. When the antenna diameter value increases, the resonance frequency decreases. The study shows the trade-off between the antenna’s target resonance frequency and the return loss level, which results in a narrow bandwidth while achieving the target operating frequency. As shown in Figure 2c, the variation of antenna pitch distance at the second stage was also monitored. The initial second-stage pitch distance (S2) was set to 3 mm and decreased to 2.5 mm and 1.5 mm sequentially. The simulation result indicates that the resonance frequency of the antenna changed towards a lower frequency, which means it has a proportional relationship with the pitch length. Looking at Figure 2d, the helix’s third-stage number of turns increased from 6.5 turns to 7.5 and 8.5 turns. This study shows the inversely proportional relationship between the central frequency of the CHA antenna and the number of turns. As we increase the value of the number of turns, the resonance frequency decreases while providing a good return loss at a lower frequency.

Based on this simulation study and according to Equation (Equation 1), we obtained the optimal design model of the non-uniform-pitch helical antenna with a diameter D=5mm and length L=16.8mm, which is approximately equal to 0.055λ. The optimized antenna exhibits resonance at 0.917 GHz with a return loss of −20.750 dB in the bandwidth range of 0.913 GHz to 0.920 GHz, as depicted in Figure 3. The antenna provides a gain of 1 dBi omnidirectional radiation pattern. The optimal model design parameters are described in Table 2, including each stage’s pitch length and number of turns.

### 2.2. Hemispherical Helical Antenna (HSHA) Design

In the previous design, the non-uniform-pitch helical antenna is well-suited for wireless surface acoustic wave (SAW) temperature sensor applications. Nevertheless, the objective of this effort is to minimize the size of the system as much as feasible. Therefore, we suggest utilizing a hemispherical helical structure to evaluate the trade-offs and drawbacks of both designs and obtain a more compact system. The hemispherical helical antenna has benefits in terms of size reduction and enhanced performance. It has a more durable and compact structure compared to other helical antenna structures like cylindrical and spherical antennae, making it more suitable for applications that demand compact designs [26,27,28]. The pitch distance (p) of the hemispherical helical antenna is constant, with *N* turns of the hemispherical shape, while $N_{t}=2N$ is the full-sphere number of turns. The resonance of the antenna itself can be controlled or adjusted by changing the value of the spherical radius (r) or diameter (D). The hemispherical helical antenna structure’s geometry in each axis can be obtained as below [28]:(2)$$x\left(n\right)=\sqrt{r^{2}-{\left(r-n\right)}^{2}}\times \cos\left(\frac{2\pi n}{p}\right)$$
(3)$$y\left(n\right)=\sqrt{r^{2}-{\left(r-n\right)}^{2}}\times \sin\left(\frac{2\pi n}{p}\right)$$
(4)z(n)=−n+r
where n∈[0,r] and $p=\frac{2\pi r}{N_{t}}=\frac{\pi r}{N}$.

As illustrated in Figure 4, the proposed design consists of a hemispherical helical antenna and a SAW sensor mounted on a coplanar waveguide (CPW) with an FR-4 substrate material, having a thickness of 1 mm and dimensions of 24 × 15 mm^2^. The system operates as described in the previous section.

The initial design features an antenna wire’s cross-section diameter of d=1mm. To fully conform to the hemisphere, 3 to 5 turns are suitable for achieving an omnidirectional radiation pattern and ensuring the wire fully winds around the spherical shape; increasing the number of turns can enhance the axial ratio bandwidth [29,30]. The diameter of the hemispherical helix is chosen as D=0.036λ, approximately 12mm. To analyze the characteristics of the Hemispherical Helical Antenna (HSHA), we first mounted the HSHA on the CPW, which has exact dimensions as the CPW used to feed the Cylindrical Helical Antenna (CHA). The simulation results, depicted in Figure 5a and noted with a black solid curve, show that the antenna system exhibits resonance at 0.592GHz with a return loss of −11.7dB. The resonance frequency of the HSHA can be controlled by physical parameters such as the number of turns (*N*) and diameter (*D*).

Following Equation (Equation 1), we shift the resonance frequency to a higher value by changing the number of turns from N1=4.75 to N2=3.75 turns. This adjustment results in resonance at approximately 0.768GHz, indicated by a red dashed curve in Figure 5a, where a significant loss is also observed. Since the number of turns reaches the minimum value for the wire to maintain a hemispherical shape, the next consideration is the antenna diameter, which is inversely proportional to the resonance frequency. By decreasing the diameter from D1=12mm to D2=10mm, we further analyze the impact on resonance. As seen in Figure 5b, the antenna exhibits resonance at 0.889GHz while providing poor matching at the operating band. The CPW length of the antenna is reduced from CPW1 in Figure 5c, with a length of 24mm, to the CPW2 as illustrated in Figure 5f, having a smaller length. Decreasing the length of a CPW resonator often leads to an increased resonance frequency [31]. So, the length reduction was studied, and the antenna system resonance shifted to a higher frequency, which was very close to the targeted value at 0.914 GHz with a CPW2 length of 15mm and providing a lower return loss, as depicted in Figure 5e.

We successfully designed two suitable antennae for wireless SAW temperature sensors as compared in Table 3, in which the antenna height and PCB dimension are described. The HSHA model provides an advantage in terms of a smaller size than the CHA model. In contrast, as shown in Figure 6a, the HSHA antenna model provides a smaller bandwidth in the desired operating band of 0.915 GHz. Additionally, the HSHA tends to have a smaller gain than the CHA model as illustrated in Figure 6b. However, both antennae achieve an omnidirectional radiation pattern appropriate for the ideal wireless SAW temperature sensor application.

## 3. Antenna Fabrication and Measurement

### 3.1. Return Loss

Photographs of the manufactured non-uniform-pitch Cylindrical Helical Antenna (CHA) and Hemispherical Helical Antenna are shown in Figure 7. To conduct experimental measurements, we needed to connect each of our fabricated antenna systems with a VNA using an SMA connector. The MS46122B VNA model was used, and this belongs to Anritsu’s ShockLine series. As seen in Figure 8, the fabricated models were studied using two types of measurement, which have the CHA and HSHA antennae only on the CPW1 and CPW2, respectively, without the SAW temperature sensor mounted on the PCB and vice versa. Without the SAW sensor, we can see that both antennae exhibit resonance at the desired operating band, which is 0.915 GHz. As with the simulation results above, the HSHA antenna can be a compact size or smaller than the CHA model, but we observed a drawback in terms of bandwidth after measuring the return loss of the antenna. The SAW sensor alone was also measured, and a small resonance at around 0.915 GHz was observed. Based on this SAW sensor’s performance, when we combined the SAW sensors with the antennae together and measured the return loss using a vector network analyzer, the measurement of the antenna with the presence of a SAW sensor was obtained. It had an initial central frequency of 0.915 GHz and could be changed by the surrounding temperature; we observed a notch occurring at around 0.914.45 GHz for CHA and 0.915 GHz for HSHA.

### 3.2. Received Power Test

The efficacy of the helical antenna was further assessed by conducting free space path loss (FSPL) and received power measurements. The free space path loss measures the attenuation of signal power as it propagates over unobstructed space. This statistic substantially facilitates the comprehension of the efficiency and scope of wireless sensing networks.

The FSPL can be calculated using the following equation [32]:(5)$$\mathrm{FSPL}=20\log_{10}\left(d\right)+20\log_{10}\left(f\right)+20\log_{10}\left(\frac{4\pi }{c}\right)$$

Here, *d* represents the distance between the transmitter and receiver, measured in (m), and *f* represents the frequency of the signal in Hz. The symbol *c* represents the velocity of light when it is traveling through a vacuum.

The receive power test involves measuring and analyzing the power received by the helical antenna. The power received by the antenna ($P_{r}$) can be determined by applying the following equation:(6)$$P_{r}=P_{t}+G_{t}+G_{r}-\mathrm{FSPL}$$
where $P_{t}$ is the transmitted power (dBm). $G_{t}$ (dBi) is the gain of the transmitting antenna. $G_{r}$ is the gain of the receiving antenna in dBi.

In this experiment, we placed the transmitting helical antenna at a constant distance of 1 m from the receiving source, which is the TX900-PB-23223 model, which had a gain of 10.0 dBi, operating from 850 MHz to 930 MHz. We quantified the received power using a spectrum analyzer. The measurements were conducted in a room, as depicted in Figure 9. The antenna was installed vertically. The CHA and HSHA were placed as DUT for the test in 3 different transmit power $P_{t}$ values, and we observed that the HSHA model provided a lower received signal level at the receiver. It is proof that the result of the simulated gain of both models is acceptable based on the comparison of the received power test data plotted in Figure 10. The comparison between the actual received power test and the calculated power is also described in Table 4. We can see that the actual measurement is around 10 dB different from the ideal calculated one due to the other loss that we did not include in the calculation.

### 3.3. Temperature Test

In this temperature measurement, we established the test configuration based on the arrangement shown in Figure 11a,b. The Device Under Test (DUT) is linked to the Vector Network Analyzer (VNA)—model Anritsu MS2038C—via the wire connector. At the same time, the temperature controller regulates the heating plate positioned approximately 2 cm below the DUT. The Digital Thermometer is positioned adjacent to the heating plate to detect and measure the temperature surrounding the testing environment. The temperature test was performed two times, initially using the non-uniform-pitch Cylindrical Helical Antenna (CHA) and subsequently with the Hemispherical Helical Antenna (HSHA). The testing of both models was conducted with temperature in a range of 30 °C to 90 °C, which is the highest possible ambient temperature that could be achieved using the test equipment.

Figure 12a,c show the measured return loss of the CHA and HSHA antenna models, respectively. On the other side, Figure 12b,d show a closer look at the antenna performances. They emphasize the notches that were formed at the central frequency of 915 MHz due to the SAW temperature sensor effect. It was discovered that both system’s resonance frequencies shifted to lower values as the temperature increased. The measured data were recorded every 10 °C, and the frequency changes based on the test environment temperature are depicted in Figure 13. Within the test range of 30 °C to 90 °C, both CHA and HSHA systems need a 400 KHz bandwidth of the operating band for the decrement of the resonance frequencies. The linear fitting curve provides the characteristic of the antenna resonance frequency decrement with temperature. The linear fitting equation of the CHA is $f_{cha}=0.0071t+914.9$ while $f_{hsha}=0.0066t+914.35$ is the linear fitting equation for the HSHA model.

## 4. Conclusions

A comparison between the proposed design of a miniaturized non-uniform-pitch Cylindrical Helical Antenna (CHA) and Hemispherical Helical Antenna (HSHA) with the other existing antenna designed for a wireless temperature sensor is presented in Table 5. The antenna’s electrical size is compared, and the gain and bandwidth of each work are also included. Besides having a compact size, the CHA antenna also exhibits 915 MHz over a bandwidth of 30 MHz, which can cover a range of 902 MHz to 932 MHz. The maximum gain value that it can achieve is 1 dBi, and it exhibits an omniradiation pattern, which is the same as the HSHA model, which achieves a smaller and more compact size and length. However, it provides a smaller gain of −2 dB and operates over a smaller bandwidth of 22 MHz, covering 904 MHz to 926 MHz.

In conclusion, we successfully designed two compact antennae suitable for wireless SAW temperature sensor applications; the antennae include a non-uniform-pitch Cylindrical Helical Antenna (CHA) and a Hemispherical Helical Antenna operating at 915 MHz with a bandwidth of 30 MHz and 22 MHz, respectively. From the CHA system design to the HSHA system model, including feeding CPW, the overall system is deduced to be compact at around 70% smaller. The temperature test was conducted from 30 °C to 90 °C using the test set up described above, and the frequency shift was such that for every 10 °C change, the SAW sensor frequency decreased by an average of 66.67 kHz. However, the system exhibited inadequate amplification, resulting in a received power that fell short of theoretical projections. Additionally, the utilization of wire presents obstacles in the manufacturing process. A new system model is being built for future development to improve resilience and enhance gain and return loss.

## Figures and Tables

**Figure 1 sensors-24-05490-f001:**
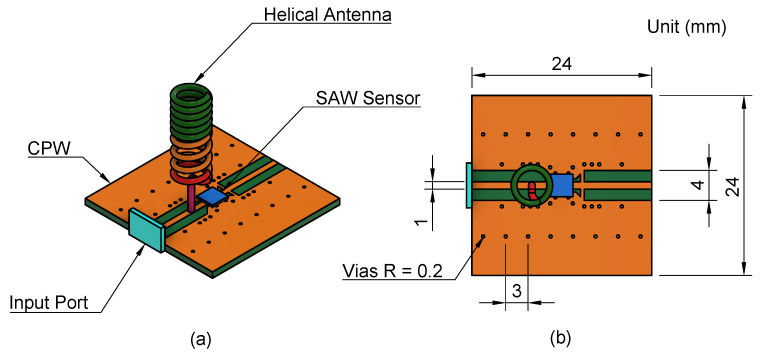
Proposed system design configuration. (**a**) Side view; (**b**) Top view and substrate dimension.

**Figure 2 sensors-24-05490-f002:**
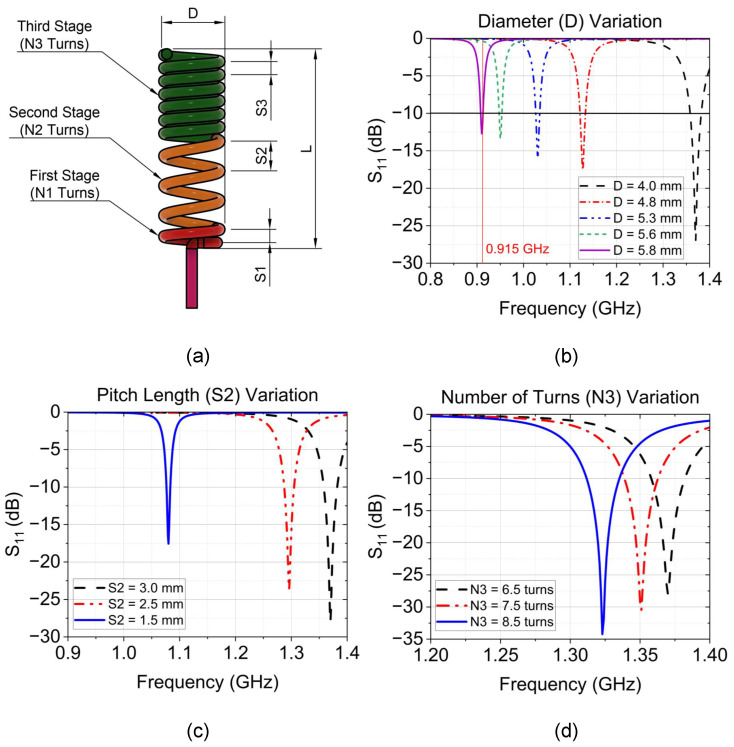
Parametric study of the CHA antenna. (**a**) Antenna dimension, (**b**) Simulation result for diameter variations, (**c**) Pitch length variation, (**d**) Change in number of turns.

**Figure 3 sensors-24-05490-f003:**
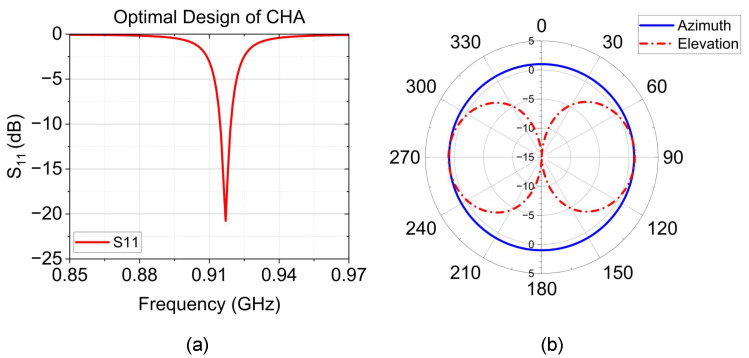
Simulation result of the optimal design of the non-uniform-pitch Cylindrical Helical Antenna (CHA). (**a**) Return loss (dB); (**b**) Radiation pattern.

**Figure 4 sensors-24-05490-f004:**
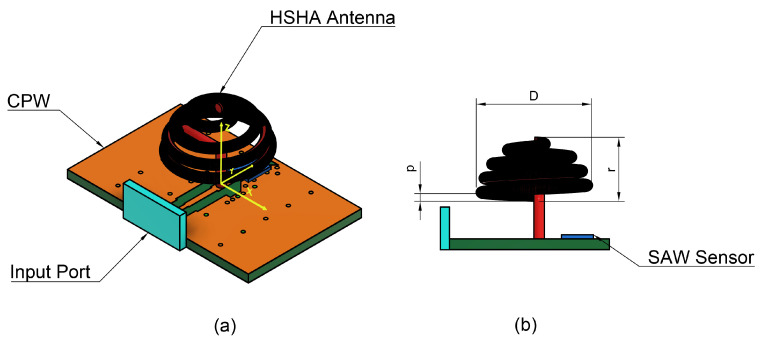
Proposed design of the Hemispherical Helical Antenna (HSHA). (**a**) Side view of system configuration; (**b**) Antenna geometry.

**Figure 5 sensors-24-05490-f005:**
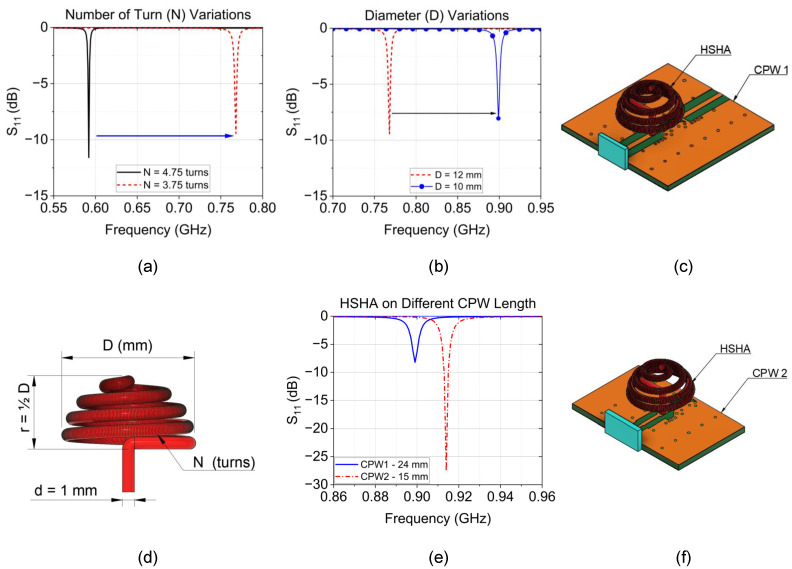
Study of Hemispherical Helical Antenna (HSHA). (**a**) Simulation result of HSHA of the number of turn variations; (**b**) Antenna diameter variations; (**c**) HSHA on CPW1 with a length of 24mm; (**d**) HSHA dimension; (**e**) HSHA performance on different lengths of CPW; (**f**) HSHA mounted on CPW2 with 15mm length.

**Figure 6 sensors-24-05490-f006:**
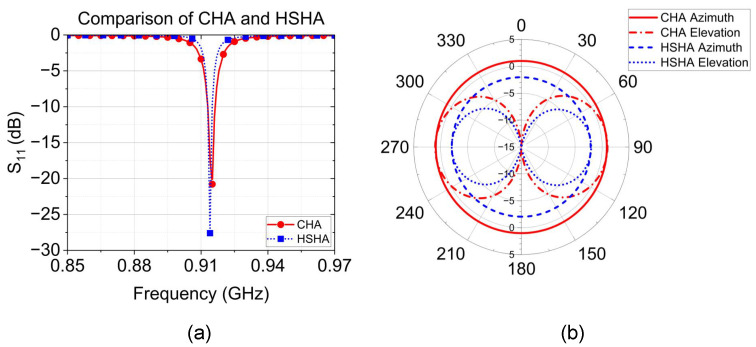
Simulation comparison of the proposed design of CHA and HSHA. (**a**) Return loss. (**b**) Radiation pattern.

**Figure 7 sensors-24-05490-f007:**
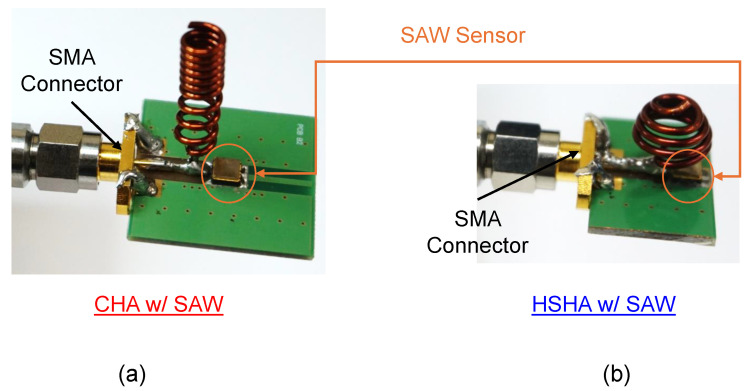
Measurement of the antenna system. (**a**) Fabricated model of CHA with SAW sensor model. (**b**) Fabricated HSHA model with SAW sensor.

**Figure 8 sensors-24-05490-f008:**
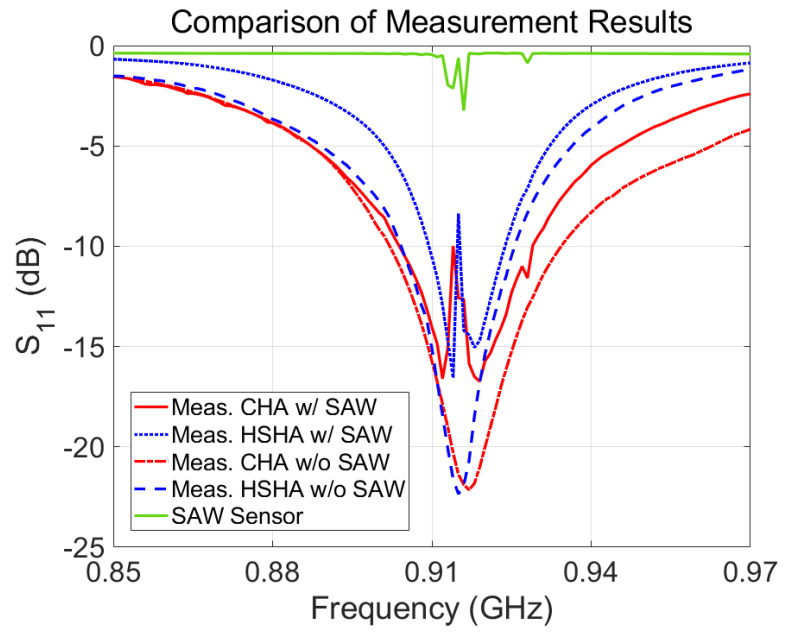
Comparison measurement result of the antenna system with the case of not having the SAW sensor mounted and with the case of the SAW sensor mounted.

**Figure 9 sensors-24-05490-f009:**
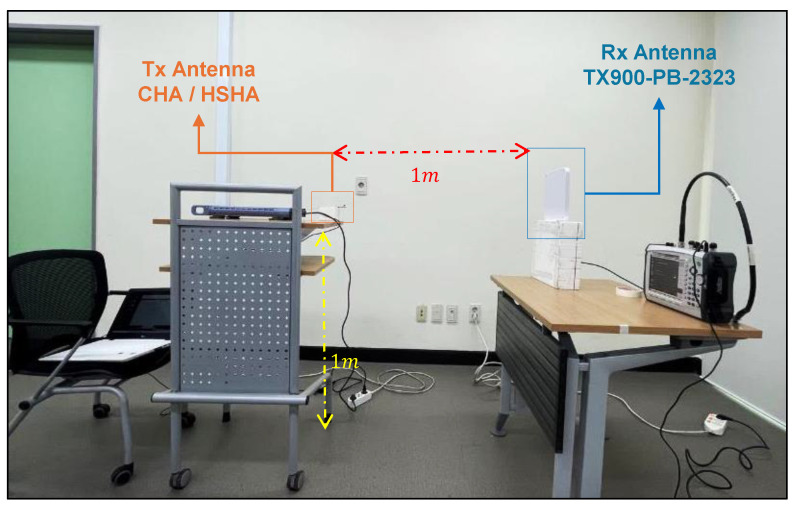
Received Power Test Setup and Configuration.

**Figure 10 sensors-24-05490-f010:**
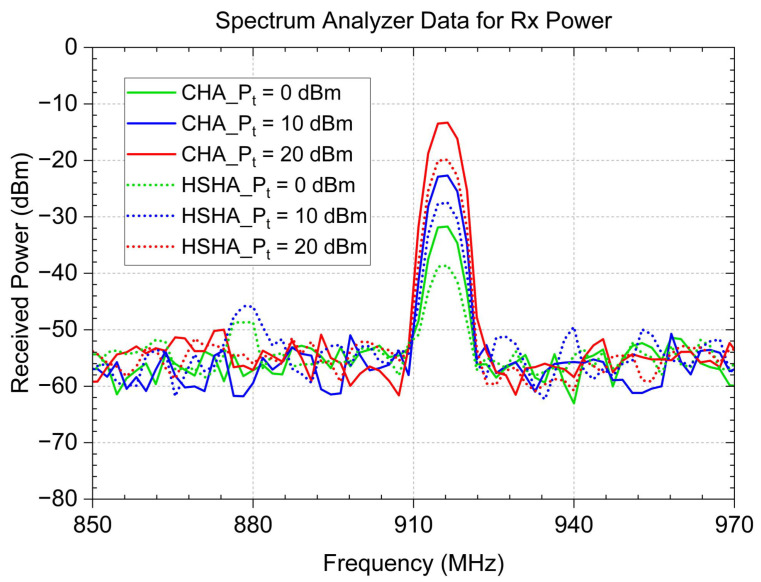
Comparison of received power measurements with different frequency and transmit power levels.

**Figure 11 sensors-24-05490-f011:**
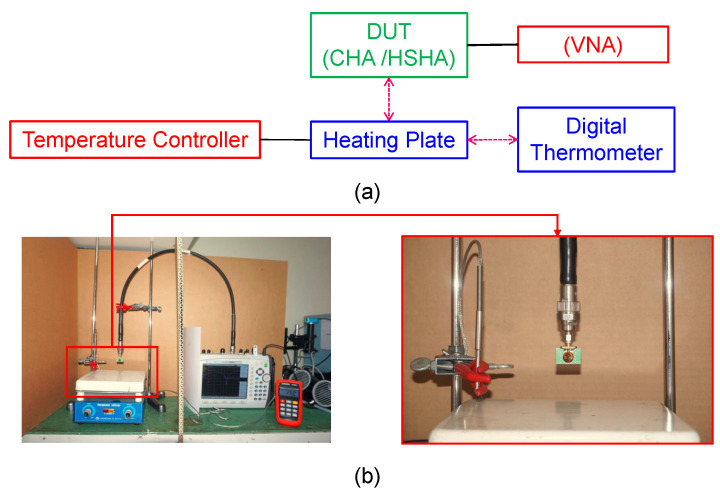
Antenna system temperature test. (**a**) Test configuration diagram; (**b**) Test setup photograph.

**Figure 12 sensors-24-05490-f012:**
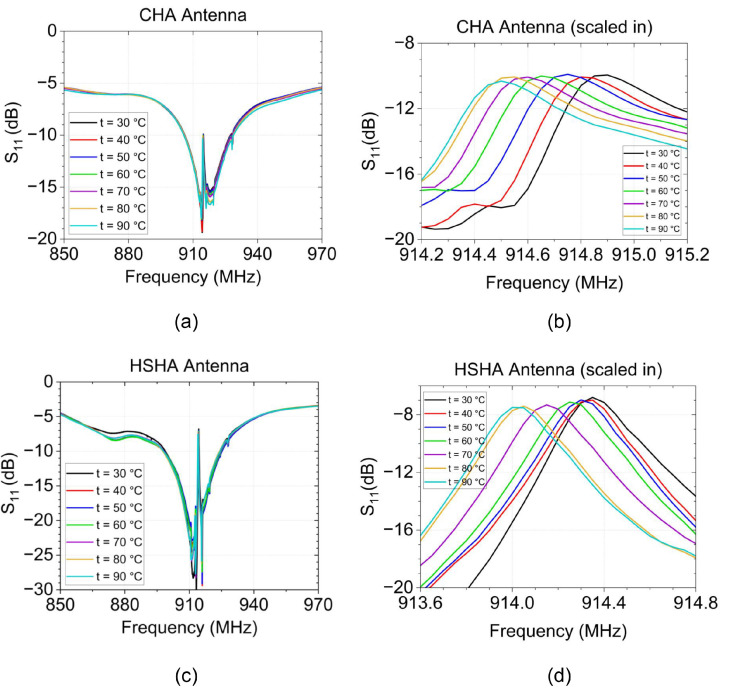
Temperature test result; changes in return loss based on temperature. (**a**) CHA antenna; (**b**) Scale-in of CHA test result; (**c**) HSHA antenna; (**d**). Scale-in of HSHA test results.

**Figure 13 sensors-24-05490-f013:**
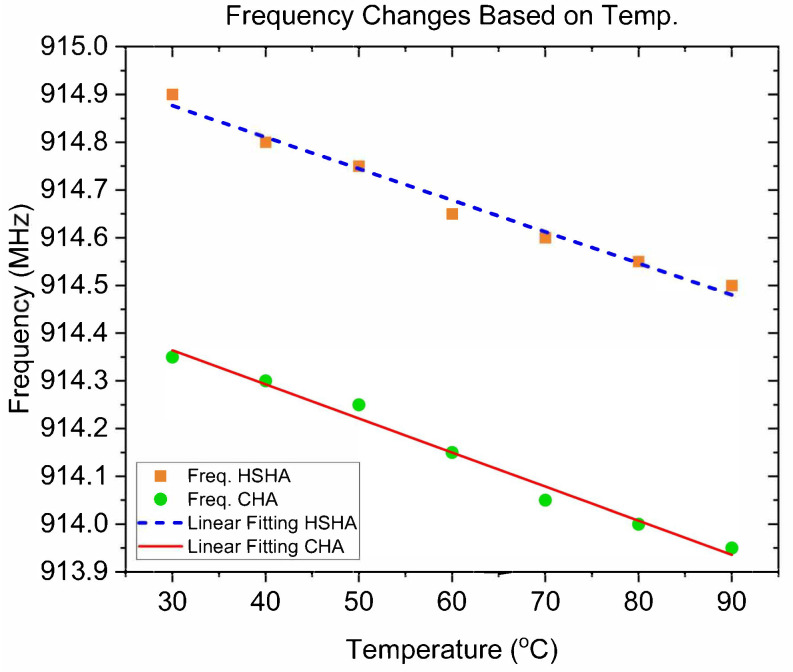
SAW sensor system frequency changes based on temperature.

**Table 1 sensors-24-05490-t001:** Design parameters of the initial proposed non-uniform Cylindrical Helical Antenna (CHA).

Helix Stage	*S* (mm)	*N* (turns)
First Stage	S1=1	N1=1.0
Second Stage	S2=3	N2=3.5
Third Stage	S3=1	N3=6.5

**Table 2 sensors-24-05490-t002:** Design parameters of the optimal design of non-uniform Cylindrical Helical Antenna (CHA).

Stage	*S* (mm)	*N* (turns)
First Stage	S1=1.2	N1=1.0
Second Stage	S2=2.5	N2=3.0
Third Stage	S3=1.1	N3=6.5

**Table 3 sensors-24-05490-t003:** Design parameters of the optimal design of non-uniform Cylindrical Helical Antenna (CHA) and Hemispherical Helical Antenna (HSHA).

Antenna Type	PCB Size (${\mathrm{mm}}^{3}$)	Antenna Height (mm)	Diameter (mm)
CHA	24×24×1	16.8	5
HSHA	24×15×1	5	10

**Table 4 sensors-24-05490-t004:** Measured and calculated Rx. power for different Tx values. Power levels at 915 MHz.

Tx. Power (dBm)	Measured Rx. Power (dBm) CHA	Measured Rx. Power (dBm) HSHA	Calculated Rx. Power (dBm) CHA	Calculated Rx. Power (dBm) HSHA
0	−31.88	−38.63	−22.36	−24.7
10	−22.88	−27.52	−12.36	−14.7
20	−13.5	−20.06	−2.36	−4.7

**Table 5 sensors-24-05490-t005:** Comparison of various existing wireless temperature sensors with antenna systems.

References	Frequency (MHz)	Bandwidth (MHz)	Ant. Length	Gain (dBi)	Ant. Volume (mm^3^)
[18]	434	20	0.029λ	2	45×45×0.2
[21]	915	30	0.110λ	2	36×20×14
[22]	915	20	0.070λ	1.4	30×30×20
**This Work**	915 (CHA)	30	0.055λ	1	24×24×21.8
	915 (HSHA)	22	0.015λ	−2	24×15×9

## Data Availability

Data are contained within the article.

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
