# Peer review of "Miniaturized Antenna Design for Wireless and Powerless Surface Acoustic Wave Temperature Sensors"

_sensors, 2024, doi:10.3390/s24175490_

Round 1

Reviewer 1 Report

Comments and Suggestions for Authors

This paper introduces the design and experimental validation of two compact helical antennas. These antennas are integral to a surface acoustic wave (SAW) sensor interrogation system optimized for operation at 915 MHz. There are some review comments as follow:

1. Figure 1 should provide detailed physical dimensions, including substrate size and relevant via dimensions.

2. In optimizing the dimensions of D in Figure b, D's range should be extended beyond 5.8 mm. Similarly, S2 should be optimized to values smaller than 1.5 mm. However, N3 in Figure 2C does not meet the frequency band requirements after optimization. Please rectify this issue and provide an explanation.

3. Figure 8 addresses the impact of SAW sensors on the antenna. How was this figure obtained, and how were the effects of the sensors on antenna performance simulated?

4. Please include radiation pattern diagrams and images from the anechoic chamber testing.

5. The comparative table should include antenna volumes, not just lengths.

6.Figures 2, 3, 5, 6, 8, 10, 12, etc., should strive to maintain consistent frequency bands.

Comments on the Quality of English Language

This paper introduces the design and experimental validation of two compact helical antennas. These antennas are integral to a surface acoustic wave (SAW) sensor interrogation system optimized for operation at 915 MHz. There are some review comments as follow:

1. Figure 1 should provide detailed physical dimensions, including substrate size and relevant via dimensions.

2. In optimizing the dimensions of D in Figure b, D's range should be extended beyond 5.8 mm. Similarly, S2 should be optimized to values smaller than 1.5 mm. However, N3 in Figure 2C does not meet the frequency band requirements after optimization. Please rectify this issue and provide an explanation.

3. Figure 8 addresses the impact of SAW sensors on the antenna. How was this figure obtained, and how were the effects of the sensors on antenna performance simulated?

4. Please include radiation pattern diagrams and images from the anechoic chamber testing.

5. The comparative table should include antenna volumes, not just lengths.

6.Figures 2, 3, 5, 6, 8, 10, 12, etc., should strive to maintain consistent frequency bands.

Author Response

Dear Reviewer, 

Thank you so much for your time and consideration in reviewing our paper and your suggestions and valuable comments are highly appreciated. 

We are writing this letter to submit our reply to the questions that you have asked. The answers to the following questions and suggestions have been described in the attachment below.

Please see the attachment below. 

Reviewer 2 Report

Comments and Suggestions for Authors

Comments to the Author

Miniaturized Antenna Design for Wireless and Powerless SAW

Temperature Sensor Comments are listed as below to help to revised the manuscript.

1:In the antenna fabrication and measurement section, the addition of statistical analysis of the results and error measurement will enhance the credibility and rigor of the study.

2:This article needs a more detailed comparison of similarities and differences with existing work, especially recent similar studies, to help highlight the unique contributions and improvements of this researches.

3:There are discrepancies between simulated and measured results of antenna performance. These differences need to be analyzed and explained in more detail.

4:Antenna efficiency at different temperatures is also one of the important concerns that is not adequately described.

Comments on the Quality of English Language

The writing is rigorous and fluent, very close to being publishable, but there is still room for improvement in the full English writing, so you can continue to refine the statements.

Author Response

(The authors gave the same response as above.)

Round 2

Reviewer 1 Report

Comments and Suggestions for Authors

The authors have responded to the review comments, but there are still some issues:

1. Is the input port in Figure 1 a wave port in HFSS? What is its size?

2. The authors have not specifically addressed Comment 2 from the previous review and have not revised the manuscript accordingly. While D is shown as the largest value in the figure, it may not be optimal. Additionally, considering size constraints, why is S2 not optimized to be smaller than 1.5 mm?

3. For Comment 3, the explanation of Figure 8 and the revised Figure 8 should be updated in the text.

4. The radiation pattern can demonstrate the antenna’s gain or reception capability in different directions and assess whether the antenna meets the design requirements. Without a radiation pattern, readers may not fully understand the antenna’s practical performance, limiting the paper’s practical value. The lack of tested radiation patterns is a significant drawback of the paper. Please find a way to include the test results.

5. It is unclear why the frequency bands in Figures 2 and 5 do not match those in the other figures. Please provide a clearer explanation.

Comments on the Quality of English Language

Minor editing of English language required.

Author Response

Dear Reviewer, 

Thank you for your time and consideration in reviewing our paper and your suggestions and valuable comments are highly appreciated. 

Please see the attachment for the response to the following comment.

Best regard,
